# Large Language Models in Action: Supporting Clinical Evaluation in an Infectious Disease Unit

**DOI:** 10.3390/healthcare13080879

**Published:** 2025-04-11

**Authors:** Giulia Lorenzoni, Anna Garbin, Gloria Brigiari, Cinzia Anna Maria Papappicco, Vinicio Manfrin, Dario Gregori

**Affiliations:** 1Unit of Biostatistics, Epidemiology and Public Health, Department of Cardiac, Thoracic, Vascular Sciences and Public Health, University of Padova, 35131 Padova, Italy; giulia.lorenzoni@unipd.it (G.L.); gloria.brigiari@ubep.unipd.it (G.B.); cinziaannamaria.papappicco@ubep.unipd.it (C.A.M.P.); 2Infectious Disease Unit, San Bortolo Hospital, ULSS 8, 36100 Vicenza, Italy; anna.garbin@studenti.unipd.it (A.G.); vinicio.manfrin@aulss8.veneto.it (V.M.)

**Keywords:** large language models, sepsis, infectious diseases, acute care

## Abstract

**Background/Objectives:** Healthcare-associated infections (HAIs), including sepsis, represent a major challenge in clinical practice owing to their impact on patient outcomes and healthcare systems. Large language models (LLMs) offer a potential solution by analyzing clinical documentation and providing guideline-based recommendations for infection management. This study aimed to evaluate the performance of LLMs in extracting and assessing clinical data for appropriateness in infection prevention and management practices of patients admitted to an infectious disease ward. **Methods:** This retrospective proof-of-concept study analyzed the clinical documentation of seven patients diagnosed with sepsis and admitted to the Infectious Disease Unit of San Bortolo Hospital, ULSS 8, in the Veneto region (Italy). The following five domains were assessed: antibiotic therapy, isolation measures, urinary catheter management, infusion line management, and pressure ulcer care. The records, written in Italian, were anonymized and paired with international guidelines to evaluate the ability of LLMs (ChatGPT-4o) to extract relevant data and determine appropriateness. **Results:** The model demonstrated strengths in antibiotic therapy, urinary catheter management, the accurate identification of indications, de-escalation timing, and removal protocols. However, errors occurred in isolation measures, with incorrect recommendations for contact precautions, and in pressure ulcer management, where non-existent lesions were identified. **Conclusions:** The findings underscore the potential of LLMs not merely as computational tools but also as valuable allies in advancing evidence-based practice and supporting healthcare professionals in delivering high-quality care.

## 1. Introduction

Infection control is one of the most critical challenges in modern healthcare systems and has significant implications for patient safety and quality of care. Healthcare-associated infections (HAIs), such as sepsis and device-related infections, are major contributors to morbidity and mortality and have considerable economic and social impacts. This issue is further exacerbated by an increase in antimicrobial resistance (AMR), which undermines the effectiveness of available treatments, making some infections difficult or even impossible to manage [1].

The World Health Organization (WHO) has identified antimicrobial resistance as one of the top ten threats to global public health. Each year, an estimated 700,000 deaths are attributed to infections caused by resistant microorganisms, with projections indicating that this number could increase to 10 million annually by 2050 if adequate interventions are not implemented [2]. Antimicrobial stewardship programs and infection control strategies are central to preventing and mitigating this growing crisis.

Managing infections in hospitalized patients requires a multidisciplinary approach. This includes early risk identification, evidence-based protocols, and the continuous monitoring of intervention outcomes [1]. Despite advancements in these areas, challenges remain, such as the heterogeneity of clinical data, incomplete documentation, and difficulties in integrating information for timely and personalized decision-making [3].

Advanced technologies, such as large language models (LLMs) powered by artificial intelligence (AI), are emerging as promising tools for improving the efficiency of infection management. An example is represented by a generative pre-trained transformer (GPT), an LLM that forms the basis for applications, such as ChatGPT, which enables users to interact with the model in real time. These systems have shown the ability to analyze large amounts of unstructured clinical data, including electronic health records, and provide recommendations that align with international guidelines [4]. Their application enables the rapid analysis of clinical documentation while supporting healthcare professionals in preventing HAIs and optimizing antimicrobial use [5].

The use of LLMs in infectious disease research has already shown promising results in areas, such as patient education for infection prevention and management [6,7], support for clinical decision-making [8], public health surveillance [9,10,11], and overall clinical management [12]. Regarding patient education, a study explored LLMs’ capability of preparing educational material about Helicobacter pylori infection with promising results [7]. On the other hand, in the context of LLM use for healthcare professional training and clinical decision-making support, another study evaluated the appropriateness of information regarding endocarditis prophylaxis for dental procedures, reporting an accuracy of 80% [8].

However, despite the transformative potential of these technologies, questions remain regarding their practical applications and ethical implications [13]. Ensuring the accuracy of the generated recommendations, addressing the need for human oversight, and mitigating biases in the underlying data are challenges that must be addressed to enable safe and effective adoption [14]. Integrating LLMs into healthcare supports clinical decision making, optimizes infection management strategies, and enhances guideline adherence. Nevertheless, the use of LLMs in healthcare represents a potential paradigm shift in infection management and prevention, contributing to improved clinical outcomes and enhanced patient safety.

The present study aimed to explore the ability of LLMs to evaluate the appropriateness of actions and prescriptions employed in the management and prevention of infections in patients hospitalized for sepsis in an infectious disease unit in the Veneto region.

## 2. Materials and Methods

This retrospective study employed anonymized records of patients admitted to the Infectious Disease Unit of San Bortolo Hospital, ULSS 8, Veneto region, Italy. The conversational implementation of the LLM GPT, ChatGPT (4o version), was employed to systematically analyze these records and provide recommendations based on evidence-based guidelines. For this study, we used a license-based version of ChatGPT, which ensured access to the most up-to-date version of the model and stability between sessions. ChatGPT was chosen because it is currently one of the most widely used and recognized LLMs in the biomedical literature. Its accessibility, generalizability, and increasing adoption in clinical research has made it a suitable choice for evaluating the feasibility of applying LLMs in real-world clinical documentation. A search on PubMed confirms this: the term ChatGPT returned 6000 results for the years 2024–2025 alone. In contrast, alternative chat-based models, such as DeepSeek, returned substantially fewer results, making their inclusion in our task both impractical and irrelevant regarding current adoption and visibility.

Specific queries were designed for each domain of interest to evaluate adherence to the guidelines. The analysis was performed sequentially, focusing on one domain at a time and covering the following aspects:Antibiotic Therapy: The appropriateness of administered antibiotic regimens was evaluated based on the type of infection and the provided guideline recommendations;Isolation Measures: The chosen isolation precautions (e.g., contact, airborne, and droplet) were assessed for compliance with the guidelines based on the patient’s clinical condition and type of infection;Pressure Ulcer Management: The dressing selection and frequency of dressing changes were analyzed according to the ulcer stage and relevant guidelines for patients with documented pressure ulcers;Urinary Catheter Management: The placement, maintenance, and removal or replacement of urinary catheters were evaluated to determine alignment with the guideline recommendations;Infusion Line Management: The placement, maintenance, and removal or replacement of infusion lines (e.g., central venous catheter (CVC), peripherally inserted central catheter (PICC)) were analyzed for adherence to clinical necessity and procedural guidelines.

Relevant clinical data were extracted from each patient’s medical record, including admission notes, discharge letters, clinical diaries, nursing diaries, and emergency department reports. The extracted information was cross-referenced with domain-specific guidelines provided by ChatGPT-4o.

ChatGPT-4o responses were structured as follows:Identification and categorization of key interventions (e.g., antibiotic type, isolation measures, and catheter use).Evaluation of the appropriateness of each intervention based on the guideline criteria.Justifications for each assessment, citing specific guideline recommendations.

The analyses were performed stepwise to ensure consistent evaluations across patients and domains. Figure 1 presents the workflow of the study.

### 2.1. Prompt Implementation

Specific prompts were designed and provided to ChatGPT-4o for each evaluation domain to standardize the analysis. Prompts were structured to guide the model in extracting relevant clinical details and assessing adherence to evidence-based guidelines. Each prompt was tailored to address key areas, such as antibiotic therapy, infusion line management, urinary catheter management, isolation precautions, and pressure ulcer care. The prompts aimed to ensure clarity, minimize ambiguity, and align the model’s responses with the clinical context.

To avoid redundancy and contextual interference between tasks, we initiated a new chat session for each clinical domain (e.g., antibiotic therapy, urinary catheter management, etc.) The model was given a structured prompt in each session, followed by task-specific international guidelines, and then each patient’s anonymized clinical record. This stepwise approach optimized the model’s focus on the task and minimized unintended carryover effects from previous prompts. Prompts are presented in the Appendix A.

### 2.2. Reference Guidelines and Gold Standard Development

ChatGPT-4o was provided with a comprehensive set of internationally recognized guidelines to ensure its recommendations adhered to evidence-based standards. The Surviving Sepsis Campaign guidelines provide critical frameworks for the early recognition and management of sepsis and septic shock [15,16]. Guidelines addressing the prevention of central line-associated bloodstream infections [17] and the latest infusion therapy standards [18] were included for infusion line management. Urinary device management was guided by recommendations to prevent catheter-associated tract infections [19,20]. Isolation precautions were evaluated using well-established infection control protocols to prevent the transmission of infectious agents [21,22]. Pressure ulcer management adheres to guidelines from the European Pressure Ulcer Advisory Panel, which provides detailed protocols for selecting appropriate dressings and replacement intervals based on ulcer stage [23]. Antimicrobial stewardship practices were evaluated using institutional program recommendations designed to optimize antibiotic use and reduce antimicrobial resistance [24].

An expert in infectious diseases first extracted the information that ChatGPT-4o was tasked with, such as details about antibiotic therapy, pressure ulcers, infusion lines, isolation measures, and urinary catheters. This process established the gold standard for evaluating ChatGPT-4o’s performance. Subsequently, the expert evaluated the appropriateness of the extracted information against the same guideline documents provided to ChatGPT-4o.

### 2.3. Study Population

To be included in the study, records must adhere to the following:Pertain to patients admitted to the Infectious Disease Unit with a documented sepsis or septic state diagnosis, had access to the emergency department before hospitalization, and were discharged from the same unit without being transferred to other units during their hospital stay.Be complete and include the following:Emergency department report;Admission notes;Nursing and medical diaries;Discharge letters.

## 3. Results

This proof-of-concept study involved seven patients’ records. All the patients were diagnosed with sepsis according to the inclusion criteria. Table 1 provides an overview of the patient’s demographic and clinical characteristics, including primary symptoms at admission, relevant medical history, and discharge diagnoses for each patient included in the analysis.

### 3.1. Antibiotic Therapy

Antibiotic therapy was assessed for appropriateness based on the clinical presentation and microbiological findings. According to the gold standard, in all seven cases, the initiation of empiric therapy was aligned with the guideline recommendations. Adjustments were made following culture results, with timely de-escalation to narrower-spectrum agents, when appropriate. For example, piperacillin/tazobactam was effectively used in cases of abdominal sepsis, followed by a transition to oral therapy as patients stabilized (Table 2). The LLM correctly identified all the antibiotic therapies administered and evaluated their appropriateness based on the guidelines provided. The assessments were accurate and consistent across all cases (no errors were detected in the antibiotic therapy extraction and appropriateness evaluation).

### 3.2. Isolation Measures

Isolation measures were implemented according to the type of infection and the transmission mode. Only standard precautions were needed for all patients because none indicated additional isolation measures.

ChatGPT attempted to extract the isolation measures applied to each patient and evaluated their appropriateness based on the clinical context and guidelines. However, inaccuracies were observed in both the extraction and evaluation processes. Specifically, the model frequently misinterpreted the clinical context and incorrectly extracted the precautions applied in five out of seven cases (it identified contact precautions, even though only standard precautions were needed and applied). In only two cases, ChatGPT-4o correctly identified and evaluated the use of standard precautions (Table 3).

“Standard” refers to general infection prevention practices that apply to all patients, including the hygiene of the hospital environment, hand hygiene, the use of personal protective equipment (PPE), safe handling and disposal of sharps, and basic principles of asepsis. The label “Contact (additional),” as extracted from ChatGPT-4o, refers to specific precautions implemented in case of suspected or confirmed infections that are spread by direct or indirect contact with the patient or the patient’s environment (e.g., use of dedicated or disposable equipment, wearing gowns and gloves before entering the room, and discarding them before leaving the room). This error likely stemmed from misinterpreting the guidelines, leading to overgeneralizing recommendations for microorganisms that did not warrant additional precautions beyond standard measures.

### 3.3. Urinary Catheter Management

According to the gold standard, three of seven patients required urinary catheterization. In all cases, the use was justified by clinical indications, such as acute urinary retention or sepsis-related monitoring. Catheters were promptly removed once necessary. The remaining four patients were appropriately managed without catheterization (Table 4). The model accurately identified catheter placement and removal practices and evaluated all actions as appropriate based on guideline recommendations. Its performance in this domain strongly aligned with best practices, with no errors recorded in the data extraction and appropriateness evaluation.

### 3.4. Infusion Line Management

Infusion line management was evaluated for all seven patients, focusing on the appropriateness of line placement, maintenance, and removal (Table 5).

According to the gold standard, the placement of lines was deemed appropriate in all cases, with maintenance and removal practices aligned with evidence-based guidelines to ensure optimal patient safety and infection control. The LLM extracted and evaluated the infusion line details but encountered some inaccuracies. ChatGPT incorrectly extracted information indicating that a CVC had been placed in two patients treated with PVC. Likely, the abbreviation “CV” (used in the documentation to indicate “urinary catheter”, in Italian “catetere vescicale”) was coded as “CVC”. As a result, the assessment of the appropriateness of infusion line management in these two cases was also inaccurate.

### 3.5. Pressure Ulcer Management

None of the seven patients had documented pressure ulcers requiring specific care (Table 6).

However, the model erroneously reported pressure ulcers in two of the seven patients without documented lesions. The cause of this error remains unclear but may be linked to ambiguous or poorly structured text in clinical or nursing records. Nevertheless, the model correctly applied the guidelines for erroneously reported pressure ulcers, selecting appropriate dressings and management practices based on ulcer staging. This suggests that while extraction errors occurred, the model demonstrated competence in aligning its recommendations with evidence-based standards for the identified scenarios.

## 4. Discussion

This proof-of-concept study aimed to evaluate the performance of LLMs in supporting the management and prevention of infections by analyzing clinical documentation and providing recommendations based on evidence-based guidelines.

There are already examples in the literature demonstrating the use of LLMs in infectious diseases, both from a public health perspective [10,11] and a more clinical standpoint [12], with promising results. To our knowledge, no previous study has evaluated the ability of LLMs to assess the appropriateness of the actions and prescriptions used in the management and prevention of healthcare-associated infections. A previous study analyzed the accuracy of responses generated by LLMs regarding infective endocarditis prophylaxis in the context of dental procedures [8]. Although GPT demonstrated significantly superior performance to the other models tested, the study was not based on real-world clinical data.

Our study contributes to a growing body of evidence. The analysis demonstrated the model’s ability to extract information from clinical documentation and assess its appropriateness according to the guidelines provided, highlighting its potential as a support tool for infection management. The model demonstrated strong potential for generating accurate and clinically relevant outputs across multiple domains. Its use in antibiotic therapy and urinary catheter management is particularly noteworthy.

However, specific errors highlight the model’s limitations when applied to specific scenarios. It frequently recommends contact precautions in isolation measures even when standard precautions are sufficient. This error likely resulted from an incorrect application of guideline recommendations for the identified microorganism, as the specific characteristics of the case did not warrant such measures. A notable issue emerged in the analysis of infusion line management. The model probably incorrectly interpreted the abbreviation “CV” (used to indicate urinary catheter in the documentation language) as “CVC” (central venous catheter), leading to references to non-existent cases. Furthermore, the model reported ulcers not documented in patient records for pressure ulcer management. The cause of this error remains unclear but may be due to challenges in processing ambiguous or poorly structured input data. Such discrepancies highlight the necessity of human oversight to validate AI-generated outputs and ensure reliability in clinical decision-making.

Using clinical documentation written in a language other than English poses an additional challenge for ChatGPT-4o. Although the model is pre-trained to operate in a multilingual environment, its primary training source is English, which remains the dominant language for the biomedical literature and for performing tasks, such as information extraction, classification, and synthesis. Nevertheless, the model performed excellently when processing input in a different language, except for a few selected cases that appeared to be associated with using acronyms. These findings underscore the importance of optimizing AI tools for multilingual environments, particularly in healthcare settings, where accuracy in understanding language-specific terms is crucial. Furthermore, the small sample size, dictated by strict inclusion and exclusion criteria, limited the diversity and complexity of the cases analyzed. As this is a proof-of-concept study, further research must include a broader range of cases and clinical scenarios. That said, the limited number of cases is consistent with the study’s exploratory nature aimed at testing the feasibility of the approach. Working with a smaller, more uniform dataset allowed us to focus on specific challenges, such as the model’s difficulty in interpreting language-specific abbreviations, which are often not directly transferable from English. However, future studies should prioritize the expansion of the dataset to validate the generalizability of the results in different clinical settings and patient profiles.

## 5. Conclusions

This study explored the ability of LLMs (ChatGPT-4o) to extract clinical information and evaluate the appropriateness of patient management based on complex and unstructured documentation. Despite the limited number of cases, the results suggest that LLMs can provide valuable support for interpreting clinical narratives, with promising performance in data extraction and clinical reasoning tasks.

By offering structured insights and enhancing the assessment of complex documentation, LLMs can bridge gaps in clinical workflows, ultimately promoting informed and timely decisions. Clearly, the present study should be considered a proof of concept given the small number of cases included in the analysis. Further research is needed to better analyze the number of cases to understand ChatGPT’s capabilities in such complex tasks and to explore its integration into real-world clinical settings.

However, these preliminary findings underscore the potential of LLMs not merely as computational tools but also as valuable allies in advancing evidence-based practice and supporting healthcare professionals in delivering high-quality care.

## Figures and Tables

**Figure 1 healthcare-13-00879-f001:**
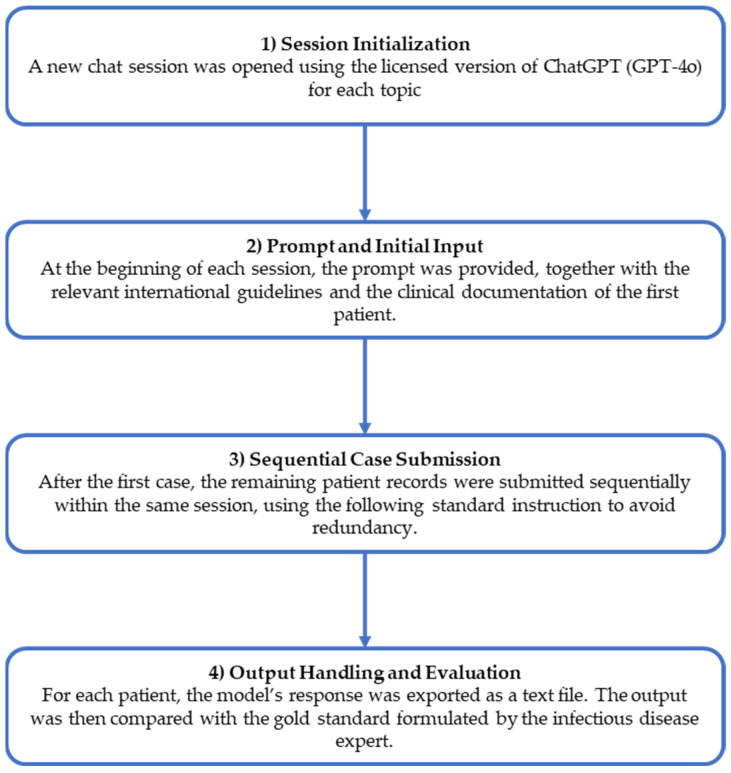
Study workflow.

**Table 1 healthcare-13-00879-t001:** Clinical characteristics of patients.

Record	Symptoms at Admission	Brief Anamnesis	Microorganism Identified	Discharge Diagnosis
1	Fever, abdominal tenderness, elevated WBC	History of abdominal pain, hypertension	*Escherichia coli*	Abdominal sepsis
2	Dysuria, fever, flank pain, leukocytosis	Recurrent urinary tract infections, diabetes mellitus	*Escherichia coli*	Acute pyelonephritis
3	Fever, dysuria, hypotension	Previous urosepsis, renal insufficiency	*Klebsiella variicola*	Urosepsis with Klebsiella infection
4	Fever, flank pain, oliguria	ESBL history, recent antibiotic use	ESBL-producing *Escherichia coli*	ESBL-associated pyelonephritis
5	Fever, neutropenia, malaise	Malignancy under chemotherapy	Empirical	Sepsis secondary to neutropenia
6	Fever, fatigue, dental pain	Dental abscess, poor oral hygiene	Empirical	Suspected endocarditis
7	Fever, tachycardia, new murmur	Known MSSA bacteremia, prosthetic valve replacement	MSSA (Methicillin-sensitive *Staphylococcus aureus*)	MSSA sepsis, suspected endocarditis

**Table 2 healthcare-13-00879-t002:** Analysis of antibiotic therapy. For each patient, the progressive numbers in the antibiotic therapy columns (actual and extracted using ChatGPT) indicate the sequence of multiple antibiotic therapies if more than one was administered.

Record	Antibiotic Therapies (Actual)	Antibiotic Therapies (Extracted by ChatGPT)	Did ChatGPT Extract the Antibiotic Data Correctly?	Was the Documented Therapy Appropriate According to the Gold Standard?	Did ChatGPT Correctly Evaluate the Appropriateness of the Extracted Therapy?
1	1 Piperacillin/Tazobactam 2 Amoxicillin/Clavulanate	1 Piperacillin/Tazobactam 2 Amoxicillin/Clavulanate	Y	Y	Y
2	1 Amoxicillin/Clavulanate	1 Amoxicillin/Clavulanate	Y	Y	Y
3	1 Piperacillin/Tazobactam 2 Trimethoprim/ Sulfamethoxazole	1 Piperacillin/Tazobactam 2 Trimethoprim/ Sulfamethoxazole	Y	Y	Y
4	1 Ceftriaxone 2 Amikacin 3 Ertapenem	1 Ceftriaxone 2 Amikacin 3 Ertapenem	Y	Y	Y
5	1 Piperacillin/Tazobactam 2 Levofloxacin	1 Piperacillin/Tazobactam 2 Levofloxacin	Y	Y	Y
6	1 Amoxicillin/Clavulanate	1 Amoxicillin/Clavulanate	Y	Y	Y
7	1 Ceftriaxone 2 Oxacillin 3 Trimethoprim/ Sulfamethoxazole	1 Ceftriaxone 2 Oxacillin 3 Trimethoprim/ Sulfamethoxazole	Y	Y	Y

Y = yes.

**Table 3 healthcare-13-00879-t003:** Analysis of isolation measures.

Record	Isolation Measure Applied (Actual)	Isolation Measures (Extracted by ChatGPT)	Did ChatGPT Extract Isolation Measures Correctly?	Were the Applied Isolation Measures Appropriate According to the Gold Standard?	Did ChatGPT Correctly Evaluate the Appropriateness of the Extracted Isolation Measures?
1	Standard	Contact (additional)	N	Y	N
2	Standard	Contact (additional)	N	Y	N
3	Standard	Contact (additional)	N	Y	N
4	Standard	Standard	Y	Y	Y
5	Standard	Contact (additional)	N	Y	N
6	Standard	Contact (additional)	N	Y	N
7	Standard	Standard	Y	Y	Y

Y = yes, N = no.

**Table 4 healthcare-13-00879-t004:** Urinary catheter management analysis.

Record	Reason for Placement (Actual)	Did ChatGPT Correctly Extract the Presence and Reason for Urinary Catheter Placement?	Was the Urinary Catheter Placement Appropriate According to the Gold Standard?	Was the Management/Removal of the Urinary Catheter Appropriate According to the Gold Standard?	Did ChatGPT Correctly Evaluate the Appropriateness of the Catheter’s Placement?	Did ChatGPT Correctly Evaluate the Appropriateness of the Catheter’s Management/Removal?
1	Significant urinary retention (600 mL)	Y	Y	Y	Y	Y
2	Not applicable	Y	NA	NA	NA	NA
3	Septic patient; monitoring urinary output	Y	Y	Y	Y	Y
4	Not applicable	Y	NA	NA	NA	NA
5	Not applicable	Y	NA	NA	NA	NA
6	Not applicable	Y	NA	NA	NA	NA
7	Acute urinary retention (bladder distention)	Y	Y	Y	Y	Y

Y = yes, NA = not applicable.

**Table 5 healthcare-13-00879-t005:** Infusion line management analysis.

Record	Line Type (Actual)	Line Type (Extracted by ChatGPT)	Did ChatGPT Correctly Extract the Type of Infusion Line?	Was the Placement of the Infusion Line Appropriate According to the Gold Standard?	Was the Management of the Infusion Line Appropriate According to the Gold Standard?	Did ChatGPT Correctly Evaluate the Appropriateness of the Line Placement?	Did ChatGPT Correctly Evaluate the Appropriateness of the Line’s Maintenance and/or Removal?
1	PVC	CVC	N	Y	Y	N	N
2	PVC	PVC	Y	Y	Y	Y	Y
3	PVC	CVC	N	Y	Y	N	N
4	PVC	PVC	Y	Y	Y	Y	Y
5	PVC	PVC	Y	Y	Y	Y	Y
6	PICC	PICC	Y	Y	Y	Y	Y
7	PVC	PVC	Y	Y	Y	Y	Y

PVC, Peripheral venous catheter; CVC, Central Venous Catheter; PICC, peripherally inserted central catheter. Y = yes, N = no.

**Table 6 healthcare-13-00879-t006:** Analysis of pressure ulcer management.

Patient	Ulcer Presence (Actual)	Ulcer Presence (Extracted by ChatGPT)	Dressing Type	Did ChatGPT Correctly Extract the Presence and Stage of Pressure Ulcers?	Was the Management of Pressure Ulcers Appropriate According to the Gold Standard?	Did ChatGPT Correctly Evaluate the Appropriateness of the Management of Pressure Ulcers?
1	None	Stage 2 Stage 3	Hydrocolloid dressing Foam with silver	N	NA	Y
2	None	None	NA	Y	NA	NA
3	None	Stage 1	Silicone foam + cream	N	NA	Y
4	None	None	NA	Y	NA	NA
5	None	None	NA	Y	NA	NA
6	None	None	NA	Y	NA	NA
7	None	None	NA	Y	NA	NA

Y = yes, N = no and NA = not applicable.

## Data Availability

The authors will make the raw data supporting this article’s conclusions available upon request.

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
