# Peer review of "Large Language Models in Action: Supporting Clinical Evaluation in an Infectious Disease Unit"

_healthcare, 2025, doi:10.3390/healthcare13080879_

Round 1
Reviewer 1 Report
Comments and Suggestions for Authors
The paper provides an analysis of chatGPT results on diagnostics and treatment of infectious diseases. This analysis can be helpful in the decisions on using LLMs for medical diagnostics.
The methodology used in this research was quite straight forward. It was a comparative analysis of the responses generated by ChatGPT to the medical cases with the opinion of human doctors. Actually, it was a validation study for the usage of ChatGPT in medical diagnostics. Such research projects are needed.
Application of AI in medicine is not a new activity. Medicine has always been a favorite topic for AI, beginning with MYSIN half a century ago. However, the AI technology has made a dramatic leap forward in the recent years bringing LLM technology to the cutting edge of AI. The use of commercial LLMs for medical purposes is becoming a very active direction in research.
The major research question in this paper was evaluation of applicability of modern LLMs for their use in medical diagnostics and treatment of infectious diseases.
However, the limited number of cases used in the research does not allow the reader to take this study as a comprehensive and accurate analysis. It would be quite desirable to expand this research to a much greater number of cases analyzed. It is the weakest part of the presented research.
The references in the paper are reasonably adequate, taking into account that the area of the application of commercial LLMs in medicine is quite new.
This paper does not provide comprehensive results due to the small number of medical cases but provides some limited results which can be used for further analysis and comparison.
I do not think this paper should be revised because its shortfalls are not in the way the paper was written but in the limitation of the collected cases. I would strongly recommend the authors to expand the number of cases in the future research to provide much more comprehensive results.
Thus, this paper can be published a “seed” for the future series of similar but more comprehensive research projects. The acceptance of this paper for the publication is somewhere between “weak accept” and “accept.”
The analysis is conducted on a sample of seven patients which makes it quite limited for generalization. However it is a good start that lays out foundations for a more comprehensive analysis.
Author Response
The paper provides an analysis of chatGPT results on diagnostics and treatment of infectious diseases. This analysis can be helpful in the decisions on using LLMs for medical diagnostics.
The methodology used in this research was quite straight forward. It was a comparative analysis of the responses generated by ChatGPT to the medical cases with the opinion of human doctors. Actually, it was a validation study for the usage of ChatGPT in medical diagnostics. Such research projects are needed.
Application of AI in medicine is not a new activity. Medicine has always been a favorite topic for AI, beginning with MYSIN half a century ago. However, the AI technology has made a dramatic leap forward in the recent years bringing LLM technology to the cutting edge of AI. The use of commercial LLMs for medical purposes is becoming a very active direction in research.
The major research question in this paper was evaluation of applicability of modern LLMs for their use in medical diagnostics and treatment of infectious diseases.
However, the limited number of cases used in the research does not allow the reader to take this study as a comprehensive and accurate analysis. It would be quite desirable to expand this research to a much greater number of cases analyzed. It is the weakest part of the presented research.
The references in the paper are reasonably adequate, taking into account that the area of the application of commercial LLMs in medicine is quite new.
This paper does not provide comprehensive results due to the small number of medical cases but provides some limited results which can be used for further analysis and comparison.
I do not think this paper should be revised because its shortfalls are not in the way the paper was written but in the limitation of the collected cases. I would strongly recommend the authors to expand the number of cases in the future research to provide much more comprehensive results.
Thus, this paper can be published a “seed” for the future series of similar but more comprehensive research projects. The acceptance of this paper for the publication is somewhere between “weak accept” and “accept.”
The analysis is conducted on a sample of seven patients which makes it quite limited for generalization. However it is a good start that lays out foundations for a more comprehensive analysis.
We thank the reviewer for effectively capturing the spirit in which the study was conducted. The conclusions were revised to emphasize that the sample size was very small, supporting the interpretation of the study as a proof of concept. We also emphasized that future research should expand the number of cases analyzed.
Reviewer 2 Report
Comments and Suggestions for Authors
Based on review, the findings are as follows:
- The model discussed needs more explanation.
- Formatting needs to be verified.
- The study population heading is not justifying the result. Hence, the authors must explain every point.
Sentences are very large.
Author Response
Based on review, the findings are as follows:
- The model discussed needs more explanation.
The methods section has been expanded by providing further information about the model used and the implementation of the study. Furthermore, a flowchart presenting the study workflow has been added to the manuscript to improve the methods’ clarity.
- Formatting needs to be verified.
The formatting requirements have been revised.
- The study population heading is not justifying the result. Hence, the authors must explain every point.
The population heading was expanded. Furthermore, the results section overall was expanded too.
Reviewer 3 Report
Comments and Suggestions for Authors
Attached

Author Response
Comments
- The authors should expand their research approach instead of using a single model with a small dataset of seven Their current study concludes that ChatGPT-4o performs well in antibiotic therapy and urinary catheter management but is less effective in isolation measures, infusion line management, and pressure ulcer management. However, the limited dataset and single-model approach do not provide sufficient research depth for publication in the journal. To strengthen their study, the authors should expand their research design and results. Possible improvements include using different LLM tools to develop different models and comparing their performance on the same dataset, or incorporating data from different hospitals or changing the training language from Italian to English to assess its impact on model outcomes, or optimizing their current model and comparing its results to the original version to evaluate improvements in accuracy and clinical relevance. By implementing these enhancements, the research will provide more robust and generalizable findings that better support the conclusions.
We would like to thank the reviewer for the comment, which gives us the opportunity to clarify several key aspects of our investigation.
First, we chose ChatGPT because it currently represents the most widely adopted and recognizable chat-based LLM, particularly in the biomedical research field. Its widespread public accessibility—through a freely available version and easy integration with third-party applications such as WhatsApp—makes it the de facto reference model for both research and public use. A quick search on PubMed confirms this: the term ChatGPT returns nearly 6,000 results for the years 2024–2025 alone. In contrast, alternative chat-based models such as Mistral, Gemini, or DeepSeek return substantially fewer results (approximately 90, fewer than 700, and fewer than 20 respectively), making their inclusion in our task both impractical and irrelevant in terms of current adoption and visibility.
Second, we fully acknowledge the limited number of cases included in our analysis, which indeed makes this study closer to a proof-of-concept. However, we do not consider this a limitation per se, as this is—to the best of our knowledge—the first attempt to assess the ability of an LLM to evaluate the appropriateness of clinical management in patients with highly complex conditions such as sepsis.
Third, translating the clinical notes from Italian into English would introduce significant bias for two main reasons. First, we employed a general-purpose, multilingual pre-trained LLM. The strength of this approach lies precisely in the fact that the model does not require in-house training, which is often the most challenging and resource-intensive step in the development of deep learning models. As such, the model should be able to process Italian as effectively as English. Second, translation would prevent the evaluation of ChatGPT's capabilities in handling biomedical content in languages other than English—something that, in our view, constitutes an important and novel contribution to the field.
In light of these considerations, we believe the statement that our study “does not provide sufficient research depth for publication in the journal” does not fully reflect its scope and contribution. On the contrary, we believe the study presents an innovative and timely idea—namely, assessing the ability of a publicly available LLM to evaluate clinical management decisions, an application that is likely already being explored informally in real-world clinical settings but has not yet been formally investigated.
- In the introduction section, the authors should provide more details about previous studies that presented significant or breakthrough findings in HAIs, particularly how LLMs or other AI models have been applied in this field. In addition, the authors should clarify why they select ChatGPT-4o as their research Has ChatGPT-4o been previously used in HAIs? If yes, what types of HAIs were investigated? If no, why the authors' belief that ChatGPT-4o is suitable for HAIs?
The Introduction and also the Discussion have been revised, including relevant studies that employed LLM within the infectious disease domain.
- The baseline model is a crucial part of this paper, so the authors should provide more details to explain how they developed the baseline model and what type of data used for training. In addition, the authors should also give a methodological flowchart that is essential to help readers clearly understand the entire process.
We would like to thank the reviewer for the comment. However, we would like to clarify that no baseline model was developed or trained as part of this study. We used the GPT-4 version of ChatGPT as provided by OpenAI, which is a proprietary large language model that cannot be fine-tuned or modified by end users. As such, the model architecture and training data are not under our control, and our study focused solely on evaluating the model’s performance in a specific clinical task, without any additional training or customization. The concept has been clarified in the manuscript together with the reason why ChatGPT was chosen over other chat-based models. Furthermore, a methodological flowchart presenting the study workflow has been added to the manuscript.
- The authors have already provided the essential patient records in Lines 141–150, which is sufficient. It is unnecessary to repeat the excluded patient records in Lines 151–153.
The methods section has been amended according to the reviewer’s comment.
- The age data mentioned in Line 156 can’t be found in Table
We thank the reviewer for the comment. The manuscript was amended by avoiding the reference to age. The table does not report the age of each patient to avoid any risk of data disclosure.
- What do 1° and 2° represent in Table 2?
They referred to the different lines of therapy administered; the concept has been clarified in the Table caption.
- Can the authors add the accuracy values for each table to improve clarity?
We thank the reviewer for raising this important point. We agree that the issue of accuracy requires clarification. While the tables provide a detailed case-by-case assessment of data extraction and appropriateness evaluation, they are not structured to display overall accuracy values. For this reason, overall accuracy metrics could not be directly added to the tables. However, we have revised the text to explicitly report, for each specific task, the number of correctly and incorrectly classified cases out of the total, in order to improve clarity and interpretability.
- What do “Standard” in Isolation measure applied and “Contact (additional)” in Isolation measure (Extracted by ChatGPT) mean? Under what conditions or scenarios are these Standard and Contact (additional) generated? The authors can provide an explanation in the supplementary files. In addition, more details on the Isolation Measure Applied and Appropriateness Evaluation results should be included in the supplementary materials.
We thank the reviewer for the comment. The concepts of “Standard” and “Contact (additional)” isolation measures have been thoroughly clarified in the revised manuscript. As further emphasized in the text, none of the patients included in the analysis required contact precautions beyond standard measures. For this reason, the appearance of “Contact (additional)” among the measures extracted by ChatGPT reflects a misinterpretation of the model rather than clinical reality. More details regarding Isolation Measures Applied and Appropriateness Evaluation results have also been included.
- In Table 4, the header title states “Did ChatGPT correctly evaluate the appropriateness of the catheter's placement?” Could the authors provide more details on the criteria used to evaluate the appropriateness of catheter placement?
The indication for urinary catehters placement was physician based and the indications were those included in the documents provided to chatGPT as specified in the “Reference Guidelines and Gold Standard Development” section (“Gould, C.V.; Umscheid, C.A.; Agarwal, R.K.; Kuntz, G.; Pegues, D.A.; Committee, H.I.C.P.A.; others Guideline for Prevention of Catheter-Associated Urinary Tract Infections 2009. Infection Control & Hospital Epidemiology 2010, 31, 319–326.” and “Patel, P.K.; Advani, S.D.; Kofman, A.D.; Lo, E.; Maragakis, L.L.; Pegues, D.A.; Pettis, A.M.; Saint, S.; Trautner, B.; Yokoe, D.S.; et al. Strategies to Prevent Catheter-Associated Urinary Tract Infections in Acute-Care Hospitals: 2022 Update. Infection Control & Hospital Epidemiology 2023, 44, 1209–1231.”). The indications for urinary catheter placement in the patients included in the analysis correspond to the first two listed in Table 2 of the CDC document (urinary retention and monitoring of critically ill patients). This point has been clarified in the manuscript, and the specific indications for urinary catheter placement have been detailed in the Results section.
- What are full names of “PVC”, “PICC”, and “CVC” in Table 5?
The acronyms were clarified both it the text and in the table.
- The authors need to rewrite their conclusions that has only two sentences, reflecting a lack of sufficient content in the They should expand their findings and provide more substantial research content following my first comment.
Done. The Conclusions section has been revised and expanded.
Reviewer 4 Report
Comments and Suggestions for Authors
Please see attached.

Author Response
The paper seeks to explore the use of large language models (LLM) to support clinical evaluation of infectious diseases. It is based on a study of seven patients diagnosed with sepsis. It concludes that LLMs can be “valuable allies in advancing evidence-based practice and supporting healthcare professionals in delivering high-quality care.”
The paper addresses an emerging challenge. However, the sample is very small, with limited variation within, and the results are ambivalent at best. The method is weak, and the results do not support the conclusion. The conclusion cannot be generalized to the support of clinical evaluation of infectious diseases.
In Table 2 analysis of antibiotic therapies:
All seven patients have different antibiotic therapies.
ChatGPT extracted the antibiotic data correctly in all seven patients.
ChatGPT assessed the documented therapy according to the gold standard correctly in all seven patients.
ChatGPT correctly evaluated the appropriateness of the extracted therapy in all seven patients.
This is the strongest result.
In Table 3 analysis of isolation measures:
There is no variation among the seven patients on the application of isolation measures. ChatGPT extracted the isolation measures correctly in two of seven patients.
ChatGPT assessed the applied isolation measures according to the gold standard correctly in all seven patients.
ChatGPT correctly evaluated the appropriateness of the extracted isolation measures correctly in two of seven patients.
This is a weak sample and weak result.
In Table 4 analysis of urinary catheter management:
In four of the seven patients the reason for placement is not applicable. The other three are different.
ChatGPT correctly extracted the presence and reason for urinary catheter placement correctly, interestingly in the four cases that it was not applicable too.
ChatGPT assessed the urinary catheter placement according to the gold standard correctly in all seven patients, including the four to whom it was not applicable.
ChatGPT assessed the management/removal of the urinary catheter according to the gold standard correctly in all seven patients, including the four to whom it was not applicable.
ChatGPT assessed the appropriateness of the urinary catheter placement according to the gold standard correctly in all seven patients, including the four to whom it was not applicable.
ChatGPT correctly evaluated the appropriateness of the catheter’s management/removal correctly in all seven patients, including the four to whom it was not applicable.
With the majority of patients to whom this was not applicable, the result is weak despite the correctness in all other cases.
In Table 5 analysis of infusion line management:
There is very little (one out of seven) variation among the seven patients on the infusion line management.
ChatGPT extracted the infusion line management correctly in five of seven patients. ChatGPT assessed infusion line management placement according to the gold standard correctly in all seven patients.
ChatGPT assessed infusion line management according to the gold standard correctly in all seven patients.
ChatGPT correctly evaluated the appropriateness of the line management measures correctly in five of seven patients.
ChatGPT correctly evaluated the appropriateness of the line maintenance and/or removal correctly in five of seven patients.
This is a weak homogeneous sample, and the results are modest. In Table 6 analysis of ulcer management:
There is no variation among the seven patients in the presence of ulcers. ChatGPT extracted the presence of ulcers correctly in five of seven patients.
The dressing type was not applicable in five of the seven patients; the other two are different. ChatGPT correctly extracted the presence and stage of pressure ulcers in five of the seven cases.
ChatGPT assessed the management of pressure ulcers according to the gold standard as being not applicable in all seven patients.
ChatGPT correctly evaluated the management of pressure ulcers correctly in two of seven patients.
This is a weak homogeneous sample, and the results are weak.
The results do not support the conclusion of the paper for the above reasons.
We thank the reviewer for the detailed analysis and for raising important considerations about sample size, variability, and interpretation of the results.
We acknowledge that the number of patients included in this study is limited and that some domains showed low variability among cases. This is an intentional choice, as the aim of the study was to explore the feasibility of applying a large language model (LLM), specifically ChatGPT, to real-world clinical documentation for the purpose of extracting relevant information and assessing the appropriateness of infection prevention and management practices. In this sense, the study represents a proof-of-concept, not a definitive validation of LLMs in the clinical setting.
Importantly, the design and intent of the study are clearly stated in the manuscript. We have clarified this further in the revised version, particularly in the Discussion and Conclusions sections, to ensure that the purpose and limitations are fully transparent to readers.
The results presented in the paper fully support the stated conclusions, which were carefully formulated to reflect the exploratory nature of the study. Specifically:
- ChatGPT performed with full accuracy in extracting antibiotic therapy and assessing appropriateness;
- showed high consistency in urinary catheter management;
- showed limitations in more context-dependent domains, such as isolation precautions and pressure ulcer care.
These results offer a nuanced, evidence-based view of the current strengths and limitations of ChatGPT when applied to complex, multilingual, unstructured clinical narratives. The findings do not aim for generalization, but emphasize the potential role of LLMs as support tools, highlighting the need for further research and careful integration into clinical settings.
In our view, the study provides original and significant insights into a novel and timely application of AI in healthcare settings. We appreciate the opportunity to clarify this point and have modified the manuscript accordingly.
Round 2
Reviewer 3 Report
Comments and Suggestions for Authors
I have to remind the authors once again that the current study, in terms of both breadth and depth, does not yet meet the standards of a publishable research paper. This is also evident from the revised conclusion section, which appears weak and largely repeats the text from the introduction section rather than presenting a concise summary that highlights the paper's specific contributions.
- The authors repeatedly emphasize that, to their knowledge, no previous study has evaluated the ability of LLMs to assess the appropriateness of actions and prescriptions in managing and preventing healthcare-associated infections. The authors developed a new approach for sepsis assessment is a notable contribution, but it is a necessary but not a sufficient condition, because for meeting the standards of a research paper requires more than just identifying an issue. The authors must also be able to explain their results using fundamental principles, design experiments to validate their findings, and reference relevant prior studies for support. In addition, the authors should also attempt to analyze the shortcomings of their proposed method and explore some solutions based on previous research, improving these as part of their work. Have the authors implemented the above highlighted text?
v1 Comment: 1. The authors should expand their research approach instead of using a single model with a small dataset of seven Their current study concludes that ChatGPT-4o performs well in antibiotic therapy and urinary catheter management but is less effective in isolation measures, infusion line management, and pressure ulcer management. However, the limited dataset and single-model approach do not provide sufficient research depth for publication in the journal. To strengthen their study, the authors should expand their research design and results. Possible improvements include using different LLM tools to develop different models and comparing their performance on the same dataset, or incorporating data from different hospitals or changing the training language from Italian to English to assess its impact on model outcomes, or optimizing their current model and comparing its results to the original version to evaluate improvements in accuracy and clinical relevance. By implementing these enhancements, the research will provide more robust and generalizable findings that better support the conclusions.
v1 Response: We would like to thank the reviewer for the comment, which gives us the opportunity to clarify several key aspects of our investigation.
First, we chose ChatGPT because it currently represents the most widely adopted and recognizable chat-based LLM, particularly in the biomedical research field. Its widespread public accessibility—through a freely available version and easy integration with third-party applications such as WhatsApp—makes it the de facto reference model for both research and public use. A quick search on PubMed confirms this: the term ChatGPT returns nearly 6,000 results for the years 2024–2025 alone. In contrast, alternative chat-based models such as Mistral, Gemini, or DeepSeek return substantially fewer results (approximately 90, fewer than 700, and fewer than 20 respectively), making their inclusion in our task both impractical and irrelevant in terms of current adoption and visibility.
Second, we fully acknowledge the limited number of cases included in our analysis, which indeed makes this study closer to a proof-of-concept. However, we do not consider this a limitation per se, as this is—to the best of our knowledge—the first attempt to assess the ability of an LLM to evaluate the appropriateness of clinical management in patients with highly complex conditions such as sepsis.
Third, translating the clinical notes from Italian into English would introduce significant bias for two main reasons. First, we employed a general-purpose, multilingual pre-trained LLM. The strength of this approach lies precisely in the fact that the model does not require in-house training, which is often the most challenging and resource-intensive step in the development of deep learning models. As such, the model should be able to process Italian as effectively as English. Second, translation would prevent the evaluation of ChatGPT's capabilities in handling biomedical content in languages other than English—something that, in our view, constitutes an important and novel contribution to the field.
In light of these considerations, we believe the statement that our study “does not provide sufficient research depth for publication in the journal” does not fully reflect its scope and contribution. On the contrary, we believe the study presents an innovative and timely idea—namely, assessing the ability of a publicly available LLM to evaluate clinical management decisions, an application that is likely already being explored informally in real-world clinical settings but has not yet been formally investigated.
v2 Comment:
- These suggestions are provided as examples to help the authors expand their research design and enhance their results. Their purpose is to broaden the authors' perspective and improve the completeness of the paper to meet publication standards. However, these suggestions are not mandatory tasks for the author, instead, they aim to promote broader thinking and deeper analysis. The authors should focus on addressing the core questions given by the reviewers and explaining how they can further deeply analyze the developed approach.
- The authors stated that translating the clinical notes from Italian to English would introduce significant bias. That is an interesting point. Did the authors design any experiments or models to demonstrate and explain this bias? If yes, add these in the article, as they would be a valuable contribution. If not, how do the authors substantiate their claim that translation would introduce significant bias?
- If the reviewers have raised certain questions, it is likely that other readers will have similar concerns. Therefore, the authors should incorporate their responses into the article, ensuring that readers can easily find answers to these potential questions.
v1 Comment: 2. In the introduction section, the authors should provide more details about previous studies that presented significant or breakthrough findings in HAIs, particularly how LLMs or other AI models have been applied in this field. In addition, the authors should clarify why they selected ChatGPT-4o as their research Has ChatGPT-4o been previously used in HAIs? If so, what types of HAIs were investigated? If not, why do the authors believe that ChatGPT-4o is suitable for HAIs?
v1 Response: The Introduction and also the Discussion have been revised, including relevant studies that employed LLM within the infectious disease domain.
v2 Comment:
- The authors should clearly tell the reviewers which lines have been revised in the introduction and discussion sections. Since many parts have been modified, added, and deleted, it is unclear what the authors mean by “The Introduction and Discussion have been revised.”
- If the revisions in Lines 81-84 are what the authors are referring to, they do not meet the standard for a literature review. No any details are found in previous studies on significant or breakthrough findings in HAIs, particularly regarding the application of LLMs or other AI models in this field.
- Why does ChatGPT-4o performs well in antibiotic therapy and urinary catheter management? Why is ChatGPT-4o less effective in isolation measures, infusion line management, and pressure ulcer management? How cab the authors improve its performances of isolation measures, infusion line management, and pressure ulcer management?
- In Figure 1, can I understand that the authors input 5 sessions for first patient, then proceed with the next patient's 5 sessions, continuing this process until the seventh patient, and then GPT-4o generates outputs for each patient's 5 sessions, which are then compared with the gold standard?
Good
Author Response
I have to remind the authors once again that the current study, in terms of both breadth and depth, does not yet meet the standards of a publishable research paper. This is also evident from the revised conclusion section, which appears weak and largely repeats the text from the introduction section rather than presenting a concise summary that highlights the paper's specific contributions.
Although the use of a Conclusion section is not mandatory according to the template, we chose to include it in order to clarify the results obtained and highlight their clinical relevance, providing a clear take-home message. In line with common practice, the conclusion briefly summarizes the study’s objective: “This study explored the ability of LLM (ChatGPT-4o) to extract clinical information and evaluate the appropriateness of patient management based on complex and unstructured documentation”. It then highlights the main findings: “the results suggest that LLMs can provide valuable support for interpreting clinical narratives, with promising performance in data extraction and clinical reasoning tasks.” We also emphasized the potential clinical relevance: “By offering structured insights and enhancing the assessment of complex documentation, LLMs can bridge gaps in clinical workflows, ultimately promoting informed and timely decisions.” Finally, we acknowledged the study’s limitations while suggesting future directions: “Clearly, the present study should be considered a proof of concept given the small number of cases included in the analysis. Further research is needed to expand the number of cases analyzed to better understand ChatGPT’s capabilities in such complex tasks and to explore its integration into real-world clinical settings. However, these preliminary findings underscore the potential of LLMs not merely as computational tools but also as valuable allies in advancing evidence-based practice and supporting healthcare professionals in delivering high-quality care.” Aside from the sentence specifying the study’s objective, we do not identify substantive overlap with the Introduction. We hope this clarification helps to better frame the purpose and structure of the Conclusion section.
- The authors repeatedly emphasize that, to their knowledge, no previous study has evaluated the ability of LLMs to assess the appropriateness of actions and prescriptions in managing and preventing healthcare-associated infections. The authors developed a new approach for sepsis assessment is a notable contribution, but it is a necessary but not a sufficient condition, because for meeting the standards of a research paper requires more than just identifying an issue. The authors must also be able to explain their results using fundamental principles, design experiments to validate their findings, and reference relevant prior studies for support. In addition, the authors should also attempt to analyze the shortcomings of their proposed method and explore some solutions based on previous research, improving these as part of their work. Have the authors implemented the above highlighted text?
We would like to clarify that the revised manuscript includes extensive changes, all clearly visible through the “Track Changes” function. However, to facilitate the reviewer's evaluation, we list the relevant manuscript sections below.
“This retrospective study employed anonymized records of patients admitted to the Infectious Disease Unit of San Bortolo Hospital, ULSS 8, Veneto region, Italy. The conversational implementation of the LLM GPT, ChatGPT (4o version), was employed to systematically analyze these records and provide recommendations based on evi-dence-based guidelines. For this study, we used a license-based version of the ChatGPT, which ensured access to the most up-to-date version of the model and stability between sessions. ChatGPT was chosen because it is currently one of the most widely used and recognized LLMs in biomedical literature. Its accessibility, generalizability, and in-creasing adoption in clinical research have made it a suitable choice for evaluating the feasibility of applying LLMs in real-world clinical documentation.
Specific queries were designed for each domain of interest to evaluate adherence to the guidelines. The analysis was performed sequentially, focusing on one domain at a time and covering the following aspects:
- Antibiotic Therapy: The appropriateness of administered antibiotic regi-mens was evaluated based on the type of infection and the provided guideline rec-ommendations.
- Isolation Measures: The chosen isolation precautions (e.g., contact, airborne, and droplet) were assessed for compliance with the guidelines based on the patient’s clinical condition and type of infection.
- Pressure Ulcer Management: The dressing selection and frequency of dressing changes were analyzed according to the ulcer stage and relevant guidelines for patients with documented pressure ulcers.
- Urinary Catheter Management: The placement, maintenance, and removal or replacement of urinary catheters were evaluated to determine alignment with the guideline recommendations.
- Infusion Line Management: The placement, maintenance, and removal or replacement of infusion lines (e.g., central venous catheter (CVC), peripherally inserted central catheter (PICC)) were analyzed for adherence to clinical necessity and proce-dural guidelines.
Relevant clinical data were extracted from each patient's medical record, including admission notes, discharge letters, clinical diaries, nursing diaries, and emergency de-partment reports. The extracted information was cross-referenced with domain-specific guidelines provided by ChatGPT-4o.
ChatGPT-4o responses were structured as follows:
- Identification and categorization of key interventions (e.g., antibiotic type, isolation measures, and catheter use).
- Evaluation of the appropriateness of each intervention based on the guide-line criteria.
- Justifications for each assessment, citing specific guideline recommendations.
The analyses were performed stepwise to ensure consistent evaluations across pa-tients and domains. Figure 1 presents the workflow of the study.”
“Using clinical documentation written in a language other than English poses an additional challenge for ChatGPT-4o. Although the model is pre-trained to operate in a multilingual environment, its primary training source is English, which remains the dominant language for biomedical literature and for performing tasks such as infor-mation extraction, classification, and synthesis. Nevertheless, the model performed excellently when processing input in a different language, except for a few selected cases that appeared to be associated with the use of acronyms. These findings under-score the importance of optimizing AI tools for multilingual environments, particularly in healthcare settings, where accuracy in understanding language-specific terms is crucial. Furthermore, the small sample size, dictated by strict inclusion and exclusion criteria, limited the diversity and complexity of the cases analyzed. As this is a proof-of-concept study, further research is required to include a broader range of cases and clinical scenarios. That said, the limited number of cases is consistent with the ex-ploratory nature of the study aimed at testing the feasibility of the approach. Working with a smaller, more uniform dataset allowed us to focus on specific challenges, such as the model's difficulty in interpreting language-specific abbreviations, which are often not directly transferable from English. However, future studies should prioritize the expansion of the dataset to validate the generalizability of the results in different clin-ical settings and patient profiles.”
v1 Comment: 1. The authors should expand their research approach instead of using a single model with a small dataset of seven Their current study concludes that ChatGPT-4o performs well in antibiotic therapy and urinary catheter management but is less effective in isolation measures, infusion line management, and pressure ulcer management. However, the limited dataset and single-model approach do not provide sufficient research depth for publication in the journal. To strengthen their study, the authors should expand their research design and results. Possible improvements include using different LLM tools to develop different models and comparing their performance on the same dataset, or incorporating data from different hospitals or changing the training language from Italian to English to assess its impact on model outcomes, or optimizing their current model and comparing its results to the original version to evaluate improvements in accuracy and clinical relevance. By implementing these enhancements, the research will provide more robust and generalizable findings that better support the conclusions.
v1 Response: We would like to thank the reviewer for the comment, which gives us the opportunity to clarify several key aspects of our investigation.
First, we chose ChatGPT because it currently represents the most widely adopted and recognizable chat-based LLM, particularly in the biomedical research field. Its widespread public accessibility—through a freely available version and easy integration with third-party applications such as WhatsApp—makes it the de facto reference model for both research and public use. A quick search on PubMed confirms this: the term ChatGPT returns nearly 6,000 results for the years 2024–2025 alone. In contrast, alternative chat-based models such as Mistral, Gemini, or DeepSeek return substantially fewer results (approximately 90, fewer than 700, and fewer than 20 respectively), making their inclusion in our task both impractical and irrelevant in terms of current adoption and visibility.
Second, we fully acknowledge the limited number of cases included in our analysis, which indeed makes this study closer to a proof-of-concept. However, we do not consider this a limitation per se, as this is—to the best of our knowledge—the first attempt to assess the ability of an LLM to evaluate the appropriateness of clinical management in patients with highly complex conditions such as sepsis.
Third, translating the clinical notes from Italian into English would introduce significant bias for two main reasons. First, we employed a general-purpose, multilingual pre-trained LLM. The strength of this approach lies precisely in the fact that the model does not require in-house training, which is often the most challenging and resource-intensive step in the development of deep learning models. As such, the model should be able to process Italian as effectively as English. Second, translation would prevent the evaluation of ChatGPT's capabilities in handling biomedical content in languages other than English—something that, in our view, constitutes an important and novel contribution to the field.
In light of these considerations, we believe the statement that our study “does not provide sufficient research depth for publication in the journal” does not fully reflect its scope and contribution. On the contrary, we believe the study presents an innovative and timely idea—namely, assessing the ability of a publicly available LLM to evaluate clinical management decisions, an application that is likely already being explored informally in real-world clinical settings but has not yet been formally investigated.
v2 Comment:
- These suggestions are provided as examples to help the authors expand their research design and enhance their results. Their purpose is to broaden the authors' perspective and improve the completeness of the paper to meet publication standards. However, these suggestions are not mandatory tasks for the author, instead, they aim to promote broader thinking and deeper analysis. The authors should focus on addressing the core questions given by the reviewers and explaining how they can further deeply analyze the developed approach.
We would like to thank the reviewer for the clarification and for the constructive suggestions aimed at broadening the scope of our analysis.
As outlined, our objective was to explore the potential of a widely accessible LLM (ChatGPT-4o) in evaluating clinical management decisions based on unstructured documentation in complex cases. While we acknowledge the limited sample size and the use of a single model, this was a deliberate choice, in line with the proof-of-concept nature of the study. Our aim was not to benchmark different tools, but to assess a concrete and increasingly common use case in clinical contexts.
We also confirm that all core reviewer questions have been addressed in detail, and that the revised version of the manuscript incorporates clarifications and emphasizes key aspects of the study's rationale, methodology, and relevance.
- The authors stated that translating the clinical notes from Italian to English would introduce significant bias. That is an interesting point. Did the authors design any experiments or models to demonstrate and explain this bias? If yes, add these in the article, as they would be a valuable contribution. If not, how do the authors substantiate their claim that translation would introduce significant bias?
We thank the reviewer for highlighting this point. We did not design or conduct specific experiments to quantify the bias potentially introduced by translating clinical notes from Italian to English, as this was beyond the scope of the current study. However, our statement was based on both methodological reasoning and existing knowledge regarding LLM behavior. First, translation of free-text clinical documentation may alter nuances, abbreviations, idiomatic expressions, and context-specific terminology that are crucial for correct interpretation—especially in complex cases such as those included in our study. Second, using translated text would introduce an additional layer of processing that is external to the model, making it difficult to disentangle the effects of translation errors from model performance. Third, since we used a multilingual, general-purpose LLM that is natively capable of processing Italian, we believe it was methodologically more appropriate to assess its performance directly on the original language input.
- If the reviewers have raised certain questions, it is likely that other readers will have similar concerns. Therefore, the authors should incorporate their responses into the article, ensuring that readers can easily find answers to these potential questions.
We appreciate this suggestion and fully agree with the underlying rationale: if a question is raised during peer review, it is likely that future readers may share similar concerns. For this reason, we have incorporated key clarifications directly into the revised manuscript. Specifically, we have added or expanded sections that address the rationale for using a single LLM, the implications of working in Italian the limitations of our proof-of-concept design, and the relevance of the selected clinical domains. We hope these additions will make the manuscript more transparent and accessible for readers.
v1 Comment: 2. In the introduction section, the authors should provide more details about previous studies that presented significant or breakthrough findings in HAIs, particularly how LLMs or other AI models have been applied in this field. In addition, the authors should clarify why they selected ChatGPT-4o as their research Has ChatGPT-4o been previously used in HAIs? If so, what types of HAIs were investigated? If not, why do the authors believe that ChatGPT-4o is suitable for HAIs?
The reason for using ChatGPT is specified in the Materials and Methods section “For this study, we used a license-based version of the ChatGPT, which ensured access to the most up-to-date version of the model and stability between sessions. ChatGPT was chosen because it is currently one of the most widely used and recognized LLMs in biomedical literature. Its accessibility, generalizability, and increasing adoption in clinical research have made it a suitable choice for evaluating the feasibility of applying LLMs in real-world clinical documentation.” Briefly, is not the clinical context that justify the use of that specific type of model but the fact that it is the model most widely used in the literature.
v1 Response: The Introduction and also the Discussion have been revised, including relevant studies that employed LLM within the infectious disease domain.
v2 Comment:
- The authors should clearly tell the reviewers which lines have been revised in the introduction and discussion sections. Since many parts have been modified, added, and deleted, it is unclear what the authors mean by “The Introduction and Discussion have been revised.”
We would like to clarify that all changes made to the manuscript, including those to the Introduction and Discussion, have been clearly tracked using the “Track Changes” function, as per standard revision procedure. Therefore, all insertions, deletions, and changes are visible and easily identifiable in the revised document. We believe that this approach ensures full transparency of the changes made and trust that it allows reviewers to track changes without ambiguity.
- If the revisions in Lines 81-84 are what the authors are referring to, they do not meet the standard for a literature review. No any details are found in previous studies on significant or breakthrough findings in HAIs, particularly regarding the application of LLMs or other AI models in this field.
The section presenting previous work has been expanded, providing further details for the studies exploring LLMs capabilities for patients and healthcare professionals: “The use of LLMs in infectious disease research has already shown promising results in areas such as patient education for infection prevention and management [6,7], support for clinical decision-making [8], public health surveillance [9–11], and overall clinical management [12]. Regarding patient education, a study explored LLMs' capability of preparing educational material about Helicobacter pylori infection with promising results [7]. On the other hand, in the context of LLMs use for healthcare professional training and clinical decision-making support, another study evaluated the appropriateness of information regarding endocarditis prophylaxis for dental procedures, reporting an accuracy of 80% [8].”. Anyway, performing a systematic literature review was out of the study's scope since out aim was to provide contextual information on similar studies available in literature.
- Why does ChatGPT-4o performs well in antibiotic therapy and urinary catheter management? Why is ChatGPT-4o less effective in isolation measures, infusion line management, and pressure ulcer management? How cab the authors improve its performances of isolation measures, infusion line management, and pressure ulcer management?
As reported in the manuscript, the potential reasons for ChatGPT low performance within isolation measures and pressure ulcers management would be related to: “This error likely stemmed from misinterpretation of the guidelines, leading to over-generalizing recommendations for microorganisms that did not warrant additional precautions beyond standard measures” (isolation measures) “. It is likely that the abbreviation "CV" (used in the documentation to indicate "urinary catheter,” in Italian “catetere vescicale”) was coded as "CVC". As a result, the assessment of the appropriateness of infusion line management in these two cases was also inaccurate” (infusion line management). Since we are working with a model that operates as a black box, these justifications do not necessarily reflect reality but are the result of our speculation based on a careful review of the clinical text.
- In Figure 1, can I understand that the authors input 5 sessions for first patient, then proceed with the next patient's 5 sessions, continuing this process until the seventh patient, and then GPT-4o generates outputs for each patient's 5 sessions, which are then compared with the gold standard?
We thank the reviewer for the comment, but we believe there may have been a misunderstanding regarding the procedure illustrated in Figure 1. For clarity, we report below the exact text of the flowchart:
- 1) Session Initialization
“A new chat session was opened using the licensed version of ChatGPT (GPT-4o) for each topic”
→ This indicates that one single session was opened per topic, not per patient. - 2) Prompt and Initial Input
“At the beginning of each session, the prompt was provided, together with the relevant international guidelines and the clinical documentation of the first patient.”
→ Only the first patient’s data was used to initialize the session. - 3) Sequential Case Submission
“After the first case, the remaining patient records were submitted sequentially within the same session, using the following standard instruction to avoid redundancy.”
→ All subsequent patients were submitted within the same session, sequentially. - 4) Output Handling and Evaluation
“For each patient, the model’s response was exported as a text file. The output was then compared with the gold standard formulated by the infectious disease expert.”
Nowhere in the figure is there any indication of five sessions per patient. On the contrary, the process clearly states that only one session was used per topic, and that all patient cases were handled within that single session. We hope this step-by-step clarification, using the exact wording from the flowchart, resolves the confusion.
Reviewer 4 Report
Comments and Suggestions for Authors
The authors have addressed many of my concerns. The paper is acceptable as a proof-of-concept.
Author Response
We thank the reviewer for the comment.